# Safety and efficacy of colchicine in COVID-19 patients: A systematic review and meta-analysis of randomized control trials

Farah Yasmin[1], Hala Najeeb[1], Abdul Moeed[1], Wardah Hassan[1], Mahima Khatri[1], Muhammad Sohaib Asghar[2], Ahmed Kunwer Naveed[1], Waqas Ullah[3], Salim Surani[4,5]*

1 Department of Internal Medicine, Dow Medical College, Dow University of Health Sciences, Karachi, Pakistan, 2 Department of Internal Medicine, Dow Ojha University Hospital, Karachi, Pakistan, 3 Thomas Jefferson University Hospitals, Philadelphia, PA, United States of America, 4 Adjunct Clinical Professor of Medicine and Pharmacology, Texas A&M University, College Station, Texas, United States of America, 5 Clinical Professor, University of Houston (Voluntary), Houston, Texas, United States of America

* srsurani@hotmail.com

**Data Availability Statement:** All relevant data are within the paper and its Supporting Information files.

## Abstract

### Background

Colchicine has been used an effective anti-inflammatory drug to treat gout diseases. Owing to its pharmacodynamic of inhibiting interleukins, it has been repurposed to target the cytokine storm post-SARS-CoV-2 invasion. The goal of this meta-analysis was to evaluate the safety profile of colchicine in COVID-19 patients using the gold-standard randomised-control trials.

### Methods

Electronic databases (Pubmed, Google Scholar, and Cochrane) were systematically searched until June 2021 and RCTs were extracted. Outcomes of interest included all-cause mortality, COVID-19 severity, mechanical ventilation, C-reactive protein and D-dimer levels. Using a random-effects model, dichotomous outcomes were pooled using odds ratios (OR) through the generic inverse variance formula while weighted mean differences were calculated using the Wan's method. P-values < 0.05 were considered statistically significant for all outcomes.

### Results

A total population of 16,048 from five RCTs were included in the analysis. Of this, 7957 were randomized to colchicine, and 8091 received standard care, with an average age of 60.67 years. Colchicine was observed to significantly reduce COVID-19 severity (OR: 0.41, 95% CI [0.22, 0.76]; p = 0.005), and CRP levels (WMD: -19.99, 95% CI [-32.09, -7.89]; p = 0.001). However, there was no significant difference in D-dimer levels (WMD: 0.31, 95% CI [-0.61, 1.23]; p = 0.51), mechanical ventilation (OR: 0.42, 95% CI [0.17, 1.03]; p = 0.06; $I^2$ = 74%) and all-cause mortality (OR: 0.98, 95% CI [0.83, 1.16]; p = 0.84) among patients receiving colchicine or standard care.

**Funding:** The authors received no specific funding for this work.

**Competing interests:** The authors have declared that no competing interests exist.

## Conclusion

Colchicine treatment decreased CRP levels and COVID-19 severity, with dimer levels, all-cause mortality and mechanical ventilation remaining seemingly unaffected. Thus, clinical trials need to be carried out that allow effective evaluation of colchicine in COVID-19 patients.

## Introduction

Coronavirus disease-19 (COVID-19) caused by the SARS-CoV-2 has piqued the interest of scientists and researchers worldwide. The disease carriers a mortality rate of 2.6% with the infection categorized into three phases namely initiation, pulmonary, and hyper-inflammatory according to increasing intensity. Each stage has its specified treatment, while antivirals and immuno-modulators are the preferred remedies of the first two stages [1–4]. Anti-inflammatory drugs are administered to curb the hyper-inflammatory state associated with increased C-reactive protein, pro-inflammatory cytokines, and chemokines in severe COVID-19 patients [5–7]. Due to its anti-inflammatory mechanisms, colchicine is suggested by experts as a potential treatment of the "cytokine storm" ensued by the aforementioned hyper-inflammation [8–10]. Randomized controlled trials (RCTs) are underway, testing the association of colchicine and COVID-19 adverse events, whereas numerous RCTs have already been carried out, COL-CORONA and RECOVERY being the larger ones. Due to the inclusion of biased observational studies and small-moderate-sized RCTs, previous meta-analyses were of low statistical power. Thus, we conducted a systematic review and meta-analysis pooling all RCTs including the larger COLCORONA and RECOVERY trials to show a holistic and comprehensive clinical evaluation of the efficacy of colchicine in COVID-19. Furthermore, this study incorporates C-reactive protein (CRP) and D-dimer levels, which play a crucial role in assessing the severity of COVID-19, as their elevated levels are associated with a poor prognosis [11,12]. Moreover, CRP being a cardinal marker of inflammation, serves as an indicator of the potency of colchicine in SARS-CoV-2 infection [13].

## Materials and methods

### Search strategy and study selection

This study was carried out along the Preferred Reporting Items for Systematic Reviews and Meta-Analyses (PRISMA) guidelines statement. [14] A comprehensive search was performed on PubMed (Medline), and Cochrane from the time of inception till June 12[th], 2021. Clinical-Trials.gov, Google Scholar, and Medrxiv were searched to identify grey literature and preprints. Medical Subject Headings (MESH terms) and keywords were used to develop a search strategy which included ['Coronavirus' OR 'COVID-19' OR 'SARS-CoV-2'] AND ['Colchicine']. S1 Table gives details of the search strategy. No filters or limitations were applied to the search results. Non-English language text was translated using the translate tool on Google. Hand-searching of review articles was performed to extract relevant studies. Two reviewers (FY and HN) independently screened titles, full texts, and abstracts of studies. Relevant studies were imported to Endnote X9 (Clarivate Analytics, US) to further remove duplicates.

### Eligibility criteria

Studies included were selected based on the following: patient population, intervention, comparison, outcomes of interest, study design, and definition.

1. Patient population: confirmed COVID-19 patients with a positive PCR test.

2. Exposure: Patients who received colchicine

3. Comparison: This includes the non-colchicine group which received the usual standard of care or placebo.

4. Outcomes of interest: COVID-19 severity, all-cause mortality, mechanical ventilation, and the change in lab parameters pre-treatment, and post-treatment (D-dimer and Creatine reactive protein levels).

5. Study design: Eligible completed randomized clinical trials were extracted to perform the meta-analysis.

Observational studies, case reports, and reviews were excluded from the meta-analysis at screening. Additionally, studies with non-human participants, children <18 years, or pregnant women were not included for analysis.

## Data extraction

Two reviewers (FY and HN) extracted studies based on title, full text, and abstract. Data were extracted according to the following fields of interest: author, year of publication, study design, sample size, age of the participants, gender, proportion of participants in colchicine and control groups, colchicine dosage, follow-up time, effect sizes, and ratios (OR, HR or RR) along with 95% confidence intervals, and the median values with their interquartile ranges of D-dimer and CRP levels. S2 Table shows the baseline characteristics of patients. Additionally, data for the proportion of patients in each group who received mechanical ventilation post-treatment witnessed disease progression to severe COVID-19 or lost their lives were documented. COVID-19 severity was defined as per the official guidelines, as (1) respiratory rate of ≥30 breaths per min; (2) oxygen saturation at rest ≤93%; (3) ratio of the partial pressure of arterial oxygen (PaO2) to a fractional concentration of oxygen inspired air (fiO2) ≤300 mmHg; (4) lung infiltrates >50% or (5) critical complication (respiratory failure, septic shock, and or multiple organ dysfunction/failure) [15].

## Study quality assessment

Two investigators (FY and HN) independently assessed the quality of clinical trials using the Revised Cochrane risk-of-bias tool for randomized trials (RoB 2) [16]. Studies were analyzed for generation of allocation sequence, randomisation of participants to exposure, selective reporting of outcomes, and missing data.

## Statistical analysis

The present meta-analysis was performed using Review Manager 5.4 (Cochrane Collaboration) software. For dichotomous outcomes, odds ratios (ORs) and relative risks (RRs) along with their 95% CI were extracted. In the case of raw data availability, ORs were calculated. Median values along with interquartile ranges were reported for continuous outcomes: D-Dimer and CRP levels. These values were converted to mean and standard error using Wan's method [17]. This meta-analysis reports a pooled effect of ORs and weighted mean differences (WMDs) using the generic-inverse variance and continuous outcome functions with a random-effects model. Each effect size was reported on a log scale, and the 95% CI was converted to standard error to normalize data distribution. All p values of < 0.05 was considered as statistically significant [17].

Heterogeneity amongst studies was assessed using the $I^2$ statistics and it is reported as percentages [18]. A value of 25% was considered low heterogeneity, 25% - 50% was moderate, while an $I^2$ value > 50% qualified for high discrepancy. Outcomes with studies that reported a high percentage of heterogeneity were subjected to sensitivity analysis to explore the effect of each study on the pooled estimate. According to Cochrane's guidelines [18], since data for at least 10 studies was unavailable, a funnel plot could not be obtained to detect publication bias.

## Results

### Study selection and characteristics

The search results yielded a total of 1152 studies, of which 989 results were screened after removing duplicates, as shown in Fig 1. 21 studies were screened for full text. Of these, seven studies were identified as case series, observational, and reviews, four studies assessed the

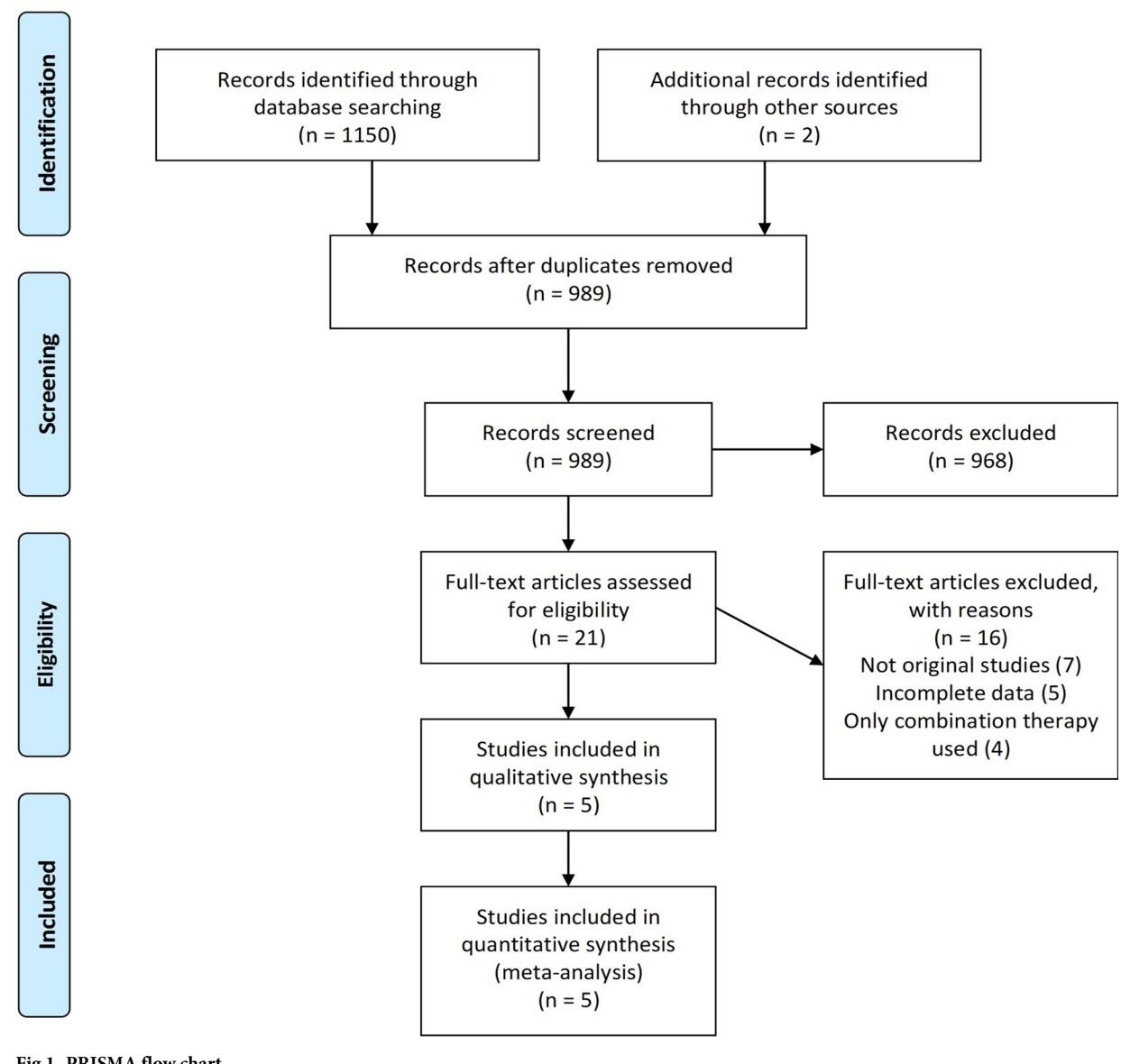

**Fig 1. PRISMA flow chart.**

**Table 1. Study characteristics of included studies.**

| Author, year | Study type | Sample size, N | Patient Population | Average age, years | | Male Sex, n (%) | Treatment duration, days | Intervention and dosage |
|---|---|---|---|---|---|---|---|---|
| | | | | Treatment | Control | | | |
| **Mareev et al, 2021** [23] | RCT | 43 | COVID-19 | 61.9 (10.6) | 59.9 (18.8) | 30 (69.8) | 13 (3.2) | 1 mg during the first 1–3 days followed by 0.5 mg/day |
| **Holby et al, 2021** [22] | RCT | 11340 | COVID-19 within 24h | 63.3 (13.8) | 63.5 (13.7) | 5609 (49.4) | 28 | N/A |
| **Tardif et al, 2021** [20] | RCT | 4488 | COVID-19 within 24 h of enrolment | 53·7 (10.4) | 54·0 (10.4) | 2067 (46.1) | 30 | 0·5 mg orally administered twice per day for the first 3 days and then once per day for 27 days |
| **Deftereos et al, 2020** [21] | RCT | 105 | Hospitalized with COVID-19 | 59.7 (4.6) | 62.7 (11.4) | 61 (58.1) | 21 | 1.5-mg loading dose followed by 0.5 mg after 60 min and maintenance doses of 0.5 mg twice daily) with standard medical treatment for as long as 3 weeks |
| **Lopes et al, 2021** [19] | RCT | 72 | Moderate-severe COVID-19 | 53.8 (17.0) | 54.7 (19.3) | 35 (48.6) | 7.0 (3.0) | 0.5 mg thrice daily for 5 days, then 0.5 mg twice daily for 5 days; if bodyweight ≥80 kg, the first dose was 1.0 mg |

effects of combination therapy while five studies were excluded because they did not report complete results of interest. Five randomized-controlled trials, Lopes et al. [19], COLCORONA [20], GRECCO [21], RECOVERY [22], and COLORIT [23] were selected for qualitative analysis. Meta-analysis was performed on a total population of 16,048 COVID-19 positive patients. Characteristics of all included RCTs are present in Table 1.

## COVID-19 severity

Three out of five studies assessed COVID-19 severity based on the above-mentioned criteria. Colchicine was shown to significantly reduce COVID-19 severity [OR = 0.41; 95% CI: 0.22–0.76, p-value 0.005; $I^2$ 0%] compared to the control as shown in Fig 2.

## All-cause mortality

All-cause mortality was reported in all five studies included. There was no significant difference in all-cause mortality [OR = 0.98; 95% CI: 0.83–1.16, p-value 0.84; $I^2$ 0%] among patients randomized to colchicine or standard care as demonstrated in Fig 3. No in-study heterogeneity was appreciated.

## Mechanical ventilation

Of the four studies included in quantitative analysis, there was no statistically significant reduction in mechanical ventilation [OR = 0.42; 95% CI: 0.17–1.03, p = 0.06] among patients

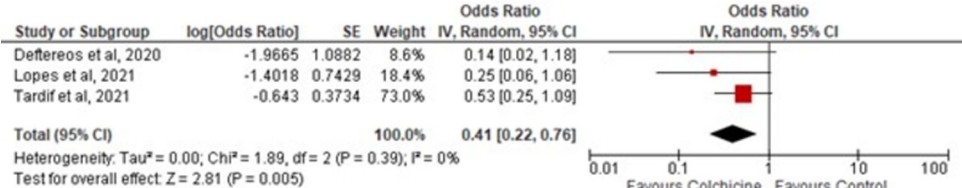

**Fig 2. Association of COVID-19 severity in colchicine users.** Red squares and their corresponding lines are the point estimates and 95% confidence intervals per each study. Black diamonds represent the pooled effect estimate.

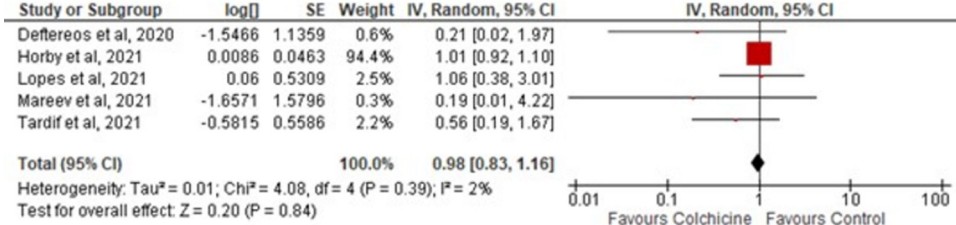

**Fig 3. Forest plot for the effect of colchicine on COVID-19 patients.** Red squares and their corresponding lines are the point estimates and 95% confidence intervals per each study. Black diamonds represent the pooled effect estimate.

randomized to colchicine vs. standard care, as shown in Fig 4. These results were subjected to a moderately-high heterogeneity ($I^2$ 74%; p-value 0.008).

**Leave-one-out sensitivity analysis.** Excluding the trials one-by-one from the pooled analysis did not reduce the in-study heterogeneity [$I^2$ 44%; p-value 0.17]. However, patients receiving colchicine had significantly lower odds of mechanical ventilation [OR = 0.26; 95% CI: 0.08–0.77, p-value 0.02] compared to standard care upon exclusion of the RECOVERY trial [22] as depicted in Fig 5.

## D-Dimer level

The COLORIT and GRECCO-19 trials were pooled to determine the effect on D-dimer inflammatory marker as shown in Fig 6. There was no significant difference in D-dimer levels among patients randomized to colchicine or standard care (p = 0.99). Moreoever, patients randomized to colchicine demonstrated a non-significant elevation of D-dimer levels post-treatment [WMD = 0.3, 95%CI:-0.61–1.23; p = 0.51].

## C-Reactive Protein (CRP) level

Colchicine was observed to significantly decrease mean CRP levels post-treatment [WMD = -19.99; 95% CI: -32.09 to -7.89, p-value 0.001; $I^2$ 98%]. On the contrary, COVID-19 patients on standard control therapy showed no reduction in mean CRP levels [WMD = 2.16; 95% CI: -7.59 to 11.92), p-value 0.66; $I^2$ 95%]. The forest plots for both analyses are shown in Fig 7.

## Risk of bias of included studies

Owing to the robust methodology of the included clinical trials, evaluation with the Revised Cochrane Risk of Bias tool (ROB 2) demonstrated a low risk of bias in all included studies. The results of the quality assessment are summarized in Fig 8.

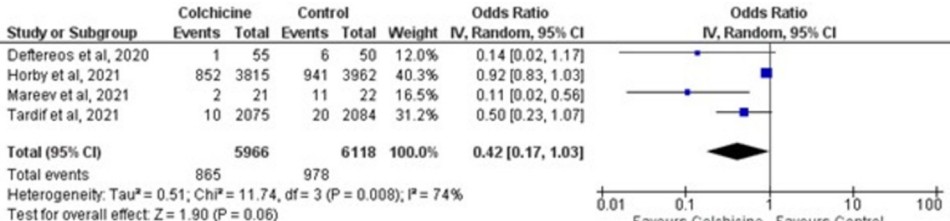

**Fig 4. Forest plot for effects of colchicine on mechanical ventilation in COVID-19 patients.** Blue squares and their corresponding lines are the point estimates and 95% confidence intervals per each study. Black diamonds represent the pooled effect estimate.

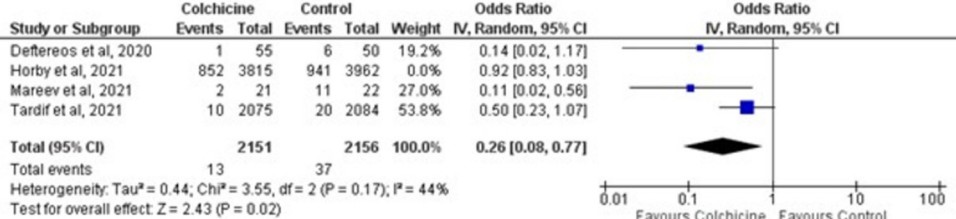

**Fig 5. Forest plot for leave-one-out sensitivity analysis of colchicine on mechanical ventilation in COVID-19 patients.** Blue squares and their corresponding lines are the point estimates and 95% confidence intervals per each study. Black diamonds represent the pooled effect estimate.

## Discussion

Our meta-analysis of five RCTs, in summary, testified to the following key findings: i) treatment with colchicine was associated with a significant reduction in the severity of COVID-19 ii) CRP levels were significantly lower in patients treated with colchicine compared to the control group. Data from multiple studies suggest an active role of Cytokine Release Syndrome (CRS) in developing ARDS and lung failure due to COVID-19 [24–26]. Based on the assumption that a 'cytokine storm' drives the progression of the disease, the anti-interleukin-6 receptor monoclonal antibody tocilizumab has been used worldwide to treat this deadly virus [27]. However, it has been suggested that there may be a 'window of opportunity' for down-modulating the immune response against the virus, without preventing its clearance [28]. In this context, colchicine, an old anti-inflammatory drug, may play a role in reversing the disease course of COVID-19.

Considerable studies have advocated that colchicine has a favorable role in treating COVID-19 [29,30]. Colchicine binds irreversibly to alpha/beta dimers and blocks the extension of microtubules. Microtubules are imperative for the migration of neutrophils towards the inflammatory focus. In addition to this, it inhibits the interaction between neutrophils and endothelial cells by altering the number and distribution of selectins, thus decreasing E-selectin-mediated adhesiveness between endothelial cells and neutrophils [31]. Studies suggest that neutrophils play an essential role in developing CRS [32]. Thus, the inhibitory effects of colchicine on neutrophil motility, adhesiveness, and chemotaxis prevent the incidence of CRS and reduce the COVID-19 severity. Likewise, it has been found that inhibition of tubulin ligands by colchicine also inhibits the replication of viruses that depends on the microtubule network [31]. At last, colchicine can also inhibit the activation of the NLRP3 inflammasome.

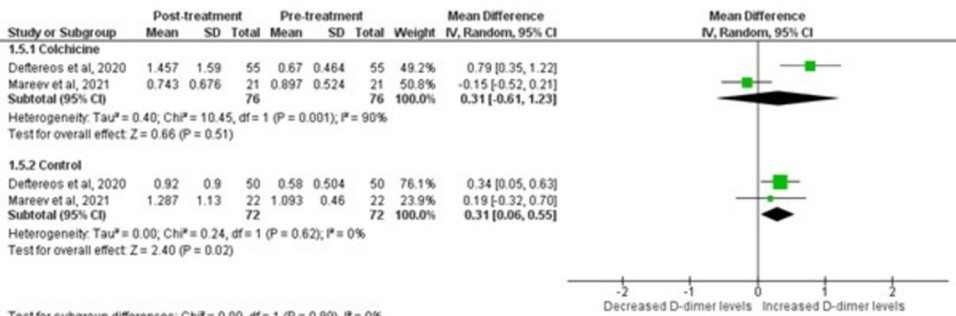

**Fig 6. Forest plot for effects of colchicine on D-Dimer in COVID-19 patients.** Green squares and their corresponding lines are the point estimates and 95% confidence intervals per each study. Black diamonds represent the pooled effect estimate.

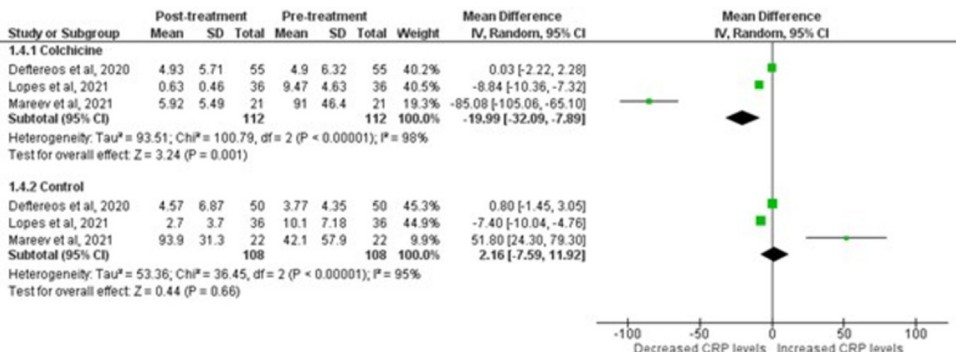

**Fig 7. Forest plot for effects of colchicine on CRP levels in COVID-19 patients.** Green squares and their corresponding lines are the point estimates and 95% confidence intervals per each study. Black diamonds represent the pooled effect estimate.

Interruption of inflammasome activation will reduce IL-1β production, which in turn prevents the induction of IL-6 and TNF-α and the recruitment of additional neutrophils and macrophages [33].

Our study shows that treatment with Colchicine was associated with a significant abatement in the severity of COVID-19 infection contrasted to the control group. This finding is coherent with the results of a meta-analysis by Timotius Ivan Hariyanto et al. [34]. It has been found that elevated levels of cytokines may cause extensive lung consolidation, which has been observed in the case of severe COVID-19 [35]. Thus, colchicine action in reducing the pro-inflammatory cytokines can alter the severity of COVID-19.

Pooling of the RCTs demonstrated that treatment with colchicine was not associated with a significant reduction in the all-cause mortality due to COVID-19 contrasted to the control group. The previous meta-analysis by Hariyanto et al. [34] exhibited a significant reduction in all-cause mortality. This difference in results could be due to the limited number of studies reporting this outcome. The inhibition of neutrophil adhesiveness and chemotaxis and replication of viruses curtails the incidence of CRS, thus forestalling the development of host autoimmune inflammatory response, which damages multiple organ systems, increasing the risk of mortality. The study by Mirko Scarsi et al. [36] endorses the idea that the prevention of the development of host autoimmune inflammatory responses causes a significant reduction in mortality rate.

Three out of five RCTs incorporated in this meta-analysis showed a significant contraction in need for mechanical ventilation with treatment with colchicine, whereas the RECOVERY trial [34] did not show a significant reduction. Reduction in the rates of COVID-19 induced lung fibrosis due to anti-inflammatory effects of colchicine [35] explains the decreased need

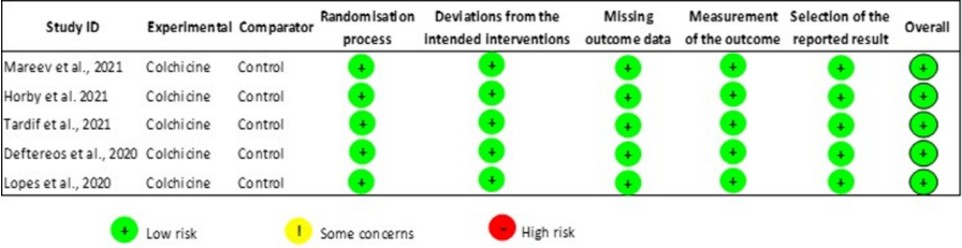

**Fig 8. Quality assessment of RCTs using revised cochrane risk-of-bias tool.**

for mechanical ventilation in the colchicine group. Owing to a limited number of studies assessing the effect of colchicine on the need for ventilation, the legitimacy of this reduced incidence cannot be counted on. Therefore, more studies should be directed towards scrutinizing the effects of Colchicine on the need for mechanical ventilation.

As per the analysis of COLORIT and GRECCO-19 trials, the colchicine group experienced slight elevation in the D-Dimer levels. A plausible reason for this outcome can be the anti-inflammatory and antithrombotic properties of colchicine on the endothelium as studied by Shiffman et al. in his study regarding Familial Mediterranean Fever [37]. D-dimer itself is a plasmin-mediated, proteolytic degradation product of cross-linked fibrin [38], which generally increases during any clinical condition; simultaneously, both clot formation and fibrinolysis occur. Direct endothelial injury in COVID-19 patients caused by inflammatory cytokines resulted in hypercoagulability, leading to increased D-dimer levels [39]. Colchicine plays a vital role in impaiing neutrophils' adhesion to the vascular endothelium; thereby, diminishing the endothelial selectin family-dependent adhesiveness [40]. Furthermore, colchicine blocked pro-inflammatory cytokines such as IL-1β, TNF-α, NF-κ, and NLRP3 inflammasome, leading to a reduction in the inflamed area and an overall decrease in the severity of the disease [41]. The individual trial of GRECCO-19 [21] depicted a decrease in the levels of risk markers of thrombotic complications, such as D-dimer and fibrinogen dropped to almost upper normal in the colchicine group; however, these decreases did not reach significance. Conversely, the COLORIT trial [23] displayed an increase in the d-dimer levels in the colchicine group versus the control group, possibly because of the activation of hypoxia-responsive signalling pathways, which further led to increased thrombogenicity [42] mainly in COVID-19 with poor prognosis [43,44]. The discrepancy in results can be explained by the involvement of only two trials in assessing the role of colchicine on D-dimer levels in COVID-19 patients and using a high loading dose of colchicine used in the COLORIT trial as compared to the GRECCO-19 trial. Therefore, in the future, large multi-center trials can display more coherent results regarding this subject.

CRP, an acute-phase reactant produced by the liver in response to inflammation, is markedly elevated in COVID-19 patients [45]. The levels of CRP fluctuate with the severity of the said disease. Therefore, it is proven to be a reliable predictor of cytokine release syndrome [46]. Extremely high CRP levels in COVID-19 patients are directly linked to poor prognosis. Our study's pooling of three RCTs delineated a significant decrease in the CRP levels in COVID-19 patients taking colchicine instead of standard therapy. This outcome highlights the role of colichine in forming tubulin-colchicine complexes leading to its anti-inflammatory effects of inhibiting chemotaxis, motility, and adhesion of neutrophils, interfering in superoxide production and inhibiting tumor necrosis factor leading to an overall decrease in cytokine storm in COVID-19 patients [47]. In the later stages of infection, intracellular viral transport and protein trafficking are mediated by microtubules; colchicine binding to the microtubule inhibits coronavirus's function and the tendency to enter the host cell [48]. A reduction in viral load is associated with decreased secreted pro-inflammatory mediators and a cumulative decrease in CRP levels.

It has been suggested that colchicine also plays a key role in destabilization and degradation of NLPR3 inflammasome leading to decreased IL-1β mediated synthesis of IL-6. In response to decreased IL-6 stimulation, the liver automatically diminishes CRP release [49], proving our study findings align with the latest review published by Reyes et al. regarding anti-inflammatory therapy for COVID-19 infection [50]. This study has several strengths. Firstly, it includes all gold-standard RCTs that largely prove the safety and efficacy of Colchicine as an anti-inflammatory medication for different inflammatory conditions, including COVID-19. We have also included the findings of recent large-scale RECOVERY trial that contributed to the

estimates in our study, and concluded that Colchicine has no significant effect on all-cause mortality and mechanical ventilation. Hence, our study expands the findings of prior literature by including contemporary trials that were missed by previous meta-analyses. The pooled estimates of prior meta-analyses were largely driven by non-randomized observational data leading to conflicting findings, the conclusion of our meta-analysis (based on only RCTs) bring consensus on the utility of Colchicine in these patients. Other reason being all included studies had some concern for risk of bias in all previous meta-analyses. Secondly, our meta-analysis comprising of only RCTs includes a summary of 16,048 COVID-19 positive patients indicating higher statistical power in comparison to previous meta-analyses. Lastly, we have demonstrated a higher methodological quality of the included trials using the standard recommended ROB-2 approach, to ensure the quality of data for inclusion of studies in our meta-analysis model. Given this, we believe our large-scale meta-analysis could serve as a benchmark against which the findings of individual studies and future trials could be compared.

## Limitations

The analysis includes a limited number of randomized controlled trials, which can lead to inadequate evidence of results with an overall increase in the risk of bias. Additionally, the duration of colchicine therapy was not entirely uniform in all the RCTs, possibly contributing to the differed results. Different dosage and the duration of the loading dose and overall length of therapy presents as a limitation of this study. Due to the low number of studies per outcome and limited available data, a sensitivity analysis was performed in place of subgroup analysis to assess high heterogeneity.

## Conclusion

Colchicine administration can be a breakthrough in the treatment of COVID-19 patients as it reduces overall disease severity. Therefore, clinicans should consider usage of colchicine in COVID-19 patients in combination or as a monotheraphy to reduce disease severity. Lastly, taking into consideration the adverse effects of colchicine in the disease prognosis, more clinical trials shall be conducted in the near future to effectively assess the efficacy and the safety of colchicine in COVID-19 patients.

## Supporting information

**S1 Checklist. PRISMA 2020 checklist.**
(DOCX)

**S1 Table. Detailed search strategy.**
(DOCX)

**S2 Table. Baseline characteristics of patients included in each study.**
(DOCX)

## Author Contributions

**Conceptualization:** Farah Yasmin, Hala Najeeb, Abdul Moeed, Wardah Hassan, Mahima Khatri, Muhammad Sohaib Asghar, Salim Surani.

**Data curation:** Farah Yasmin, Hala Najeeb, Abdul Moeed, Mahima Khatri.

**Formal analysis:** Abdul Moeed, Muhammad Sohaib Asghar, Waqas Ullah, Salim Surani.

**Investigation:** Farah Yasmin, Hala Najeeb, Abdul Moeed, Wardah Hassan, Mahima Khatri, Ahmed Kunwer Naveed, Salim Surani.

**Methodology:** Farah Yasmin, Hala Najeeb, Abdul Moeed, Salim Surani.

**Project administration:** Farah Yasmin, Waqas Ullah, Salim Surani.

**Resources:** Farah Yasmin.

**Software:** Hala Najeeb, Abdul Moeed, Muhammad Sohaib Asghar, Salim Surani.

**Supervision:** Farah Yasmin, Muhammad Sohaib Asghar, Waqas Ullah, Salim Surani.

**Validation:** Farah Yasmin, Hala Najeeb, Wardah Hassan, Muhammad Sohaib Asghar, Waqas Ullah, Salim Surani.

**Visualization:** Farah Yasmin, Hala Najeeb, Wardah Hassan, Muhammad Sohaib Asghar, Waqas Ullah, Salim Surani.

**Writing – original draft:** Farah Yasmin, Hala Najeeb, Abdul Moeed, Wardah Hassan, Mahima Khatri, Ahmed Kunwer Naveed.

**Writing – review & editing:** Farah Yasmin, Muhammad Sohaib Asghar, Waqas Ullah, Salim Surani.

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
