## [Editor Report · Decision Letter 0]

7 Dec 2021

PONE-D-21-32538Safety and efficacy of colchicine in COVID-19 patients: A systematic review and meta-analysis of randomized control trialsPLOS ONE

Dear Dr. Surani,

Thank you for submitting your manuscript to PLOS ONE. After careful consideration, we feel that it has merit but does not fully meet PLOS ONE’s publication criteria as it currently stands. Therefore, we invite you to submit a revised version of the manuscript that addresses the points raised during the review process.

We look forward to receiving your revised manuscript.

Kind regards,

Tariq Jamal Siddiqi

Academic Editor

PLOS ONE

Journal Requirements:

Additional Editor Comments:

Surani et al. conducted a meta-analysis on, “Safety and efficacy of colchicine in COVID-19 patients”, in which they reported that colchicine treatment decreased CRP levels and COVID-19 severity, with dimer levels, all-cause mortality and mechanical ventilation remaining seemingly unaffected. In my opinion, this study can be improved by incorporating the following points:

1. The strengths of the study are not clear, and it should be highlighted more as to what this study adds to the medical literature and how they address the current gap

2. The limitations may be somewhat enhanced further by highlighting additional limitations that resulted in study heterogeneity or why subgroup analysis was not performed.

3. Overall the quality of discussion is commendable, but it needs to proofread for example the capitalization error in line 281.

4. It can enhance the quality of results further if the number of studies that were included for mechanical ventilation and all-cause mortality was mentioned at the beginning of the paragraph.
---

## [Author Response · Author response to Decision Letter 0]

3 Mar 2022

January 10th, 2022

Editor-in-Chief 

PLOS-One

Dear Editor,

We are pleased to submit our revised manuscript entitled ‘Safety and efficacy of colchicine in COVID-19 patients: A systematic review and meta-analysis of randomized control trials’ for consideration for publication in PLOS-One.

We appreciate the comments we received from the editors and reviewers, all of which have been addressed in this revised version of the paper. Please find below the editor’s and reviewer’s comments in bold and the author responses. We further ensure that the material has not and will not be offered elsewhere for possible publication, as it is under consideration.

We hope that these changes will make the manuscript competitive for publication in the journal.

Salim Surani, MD, MPH, MSHM, FACP, FCCP, FAASM

Adjunct Clinical Professor of Medicine and Pharmacology, 

Texas A&M University, USA.

Clinical Professor, University of Houston (Voluntary

Email: salimsurani@gmail.com

You state: The strengths of the study are not clear, and it should be highlighted more as to what this study adds to the medical literature and how they address the current gap.

Response: Thank you so much for pointing this out. We have now edited the manuscript to include the following details:

This study has several strengths. Firstly, it includes all gold-standard RCTs that largely prove the safety and efficacy of Colchicine as an anti-inflammatory medication for different inflammatory conditions, including COVID-19. We have also included the findings of recent large-scale RECOVERY trial that contributed to the estimates in our study, and concluded that Colchicine has no significant effect on all-cause mortality and mechanical ventilation. Hence, our study expands the findings of prior literature by including contemporary trials that were missed by previous meta-analyses. The pooled estimates of prior meta-analyses were largely driven by non-randomized observational data leading to conflicting findings, the conclusion of our meta-analysis (based on only RCTs) bring consensus on the utility of Colchicine in these patients. Other reason being all included studies had some concern for risk of bias in all previous meta-analyses. Secondly, our meta-analysis comprising of only RCTs includes a summary of 16,048 COVID-19 positive patients indicating higher statistical power in comparison to previous meta-analyses. Lastly, we have demonstrated a higher methodological quality of the included trials using the standard recommended ROB-2 approach, to ensure the quality of data for inclusion of studies in our meta-analysis model. Given this, we believe our large-scale meta-analysis could serve as a benchmark against which the findings of individual studies and future trials could be compared. 

You state: The limitations may be somewhat enhanced further by highlighting additional limitations that resulted in study heterogeneity or why subgroup analysis was not performed.

Response: Thank you so much for pointing this out. We have now edited the manuscript to include the following details:

Different dosage and the duration of the loading dose and overall length of therapy presents as a limitation of this study. Due to the low number of studies per outcome and limited available data, a sensitivity analysis was performed in place of subgroup analysis to assess high heterogeneity.

You state: Overall the quality of discussion is commendable, but it needs to proofread for example the capitalization error in line 281.

Response: Thank you for pointing this out. The manuscript has been proof-read and corrected for grammatical and spelling errors.

You state: It can enhance the quality of results further if the number of studies that were included for mechanical ventilation and all-cause mortality was mentioned at the beginning of the paragraph.

Response: Thank you so much for pointing this out. We have now edited the manuscript to include the following details:

All-cause mortality was reported in all five studies included.

And

Of the four studies included in quantitative analysis, there was no statistically significant reduction in mechanical ventilation.

---

## [Decision Letter · Decision Letter 1]

17 Mar 2022

Safety and efficacy of colchicine in COVID-19 patients: A systematic review and meta-analysis of randomized control trials

PONE-D-21-32538R1

Dear Dr. Surani,

We’re pleased to inform you that your manuscript has been judged scientifically suitable for publication and will be formally accepted for publication once it meets all outstanding technical requirements.

Kind regards,

Tariq Jamal Siddiqi

Academic Editor

PLOS ONE

Additional Editor Comments (optional):

Reviewers' comments:

Reviewer's Responses to Questions

**Comments to the Author**

1. If the authors have adequately addressed your comments raised in a previous round of review and you feel that this manuscript is now acceptable for publication, you may indicate that here to bypass the “Comments to the Author” section, enter your conflict of interest statement in the “Confidential to Editor” section, and submit your "Accept" recommendation.

Reviewer #1: All comments have been addressed

2. Is the manuscript technically sound, and do the data support the conclusions?

Reviewer #1: Yes

3. Has the statistical analysis been performed appropriately and rigorously? 

Reviewer #1: Yes

4. Have the authors made all data underlying the findings in their manuscript fully available?

Reviewer #1: Yes

5. Is the manuscript presented in an intelligible fashion and written in standard English?

Reviewer #1: Yes

6. Review Comments to the Author

Reviewer #1: (No Response)

7. PLOS authors have the option to publish the peer review history of their article (what does this mean?). If published, this will include your full peer review and any attached files.

Reviewer #1: No

---

## [Editor Report · Acceptance letter]

28 Mar 2022

PONE-D-21-32538R1 

Safety and efficacy of colchicine in COVID-19 patients: A systematic review and meta-analysis of randomized control trials 

Dear Dr. Surani:

I'm pleased to inform you that your manuscript has been deemed suitable for publication in PLOS ONE. Congratulations! Your manuscript is now with our production department. 

Kind regards, 

on behalf of

Dr. Tariq Jamal Siddiqi 

Academic Editor

PLOS ONE